# Pressure vessel-oriented visual inspection method based on deep learning

**Pu Liao**, **Liu Guixiong** *

School of Mechanical and Automotive Engineering, South China University of Technology, Guangzhou, China

* megxliu@scut.edu.cn

## Abstract

The detection of surface parameters of pressure vessel welds guarantees safe operation. To address the problems of low efficiency and poor accuracy of traditional manual inspection methods, a method for welding morphological parameters combined with vision and structured light is proposed in this study. First, a feature point extraction algorithm for weld parameters based on deep convolution was proposed. An accurate extraction method of weld image feature point coordinates was designed based on the combination of the loss function via seam undercut feature recognition and weld feature point extraction network structure. Second, a training data enhancement method based on the third-order non-uniform rational B-spline (**NURBS**) curve was proposed to reduce the amount of data collection for training. Finally, a pressure vessel measurement device was designed, and the feature point extraction performance of the deep network and common feature point extraction networks, DeepLabCut and HR-net, proposed in this study were compared to analyze the theoretical accuracy of the surface parameter measurement. The results indicated that the theoretical accuracy of the parameter measurements was within 0.065 mm.

## Introduction

The butt welds of pressure vessels A and B are important factors that affect stress. The weld surface consists of four parameters: weld width, reinforcement, undercut, and misalignment, as specified in the relevant standards. These parameters directly reflect the stress concentration at the welding position, and their measurement is an important for evaluation the welding quality [1,2]. Currently, relatively mature parameter measurement methods are manually completed using a magnifying glass, weld inspection ruler, angle ruler, and other tools. However, these methods exhibit low accuracy, low efficiency, large workload, and high labor intensity. Given the good characteristics of machine vision inspection methods, there have been various degrees of research and applications in welding process inspection or welding seam surface parameters.

Based on the different imaging light sources, machine vision weld inspection methods can be classified into two branches: passive and active visual weld inspection. The passive vision welding seam detection method involves imaging in a natural light environment, using

**Data Availability Statement:** The data underlying the results presented in the study are available from https://gitee.com/Meliao/CAD.

**Funding:** This work was supported by the Science and Technology Plan Project of the State Administration for Market Regulation, grant

number 2019MK143. The funders had no role in study design, data collection and analysis, decision to publish, or preparation of the manuscript.

**Competing interests:** The authors have declared that no competing interests exist.

clustering, template matching, Hough transformation, and other algorithms to identify and locate the weld defect information (undercut, wrong edge, etc.) in the image. For example, Kumar used a BP neural network to identify and classify MIG butt welding surface defects according to the standard EN25817 [3]. Apostolos (2015) used the Sobel edge detection algorithm to obtain the undercut edge feature of the weld image and matched the undercut feature template with the feature extracted from the edge of the image, which was detected to realize weld undercut feature recognition [4]. Ding (2016) proposed the use of the circular Hough transformation to locate weld images after edge extraction for determining the location of weld defects [5]. Passive visual weld inspection method has been mainly used in the field of weld defect recognition and classification. However, owing to the poor reflection ability of the weld parameters, it is impossible to calculate the specific values of the weld size parameters in natural-light weld imaging.

Active vision imaging refers to imaging under an artificial light source (including ultraviolet light, infrared light, and laser.). The artificial light source applied to weld inspection is the most structured laser, which can be used to inspect and capture the laser line on the welded surface. The parameter information of the curved welded surface can be expressed as the feature points of the curve. [6,7]. The extraction of the feature points of the welding parameter measurement from the welding seam laser line is important for active vision welding seam measurements [8]. Current mainstream feature point determination methods include the feature point extraction method based on curve slope analysis [9], corner detection method curve inflection point [10], and curve processing and fitting [11]. The feature point extraction methods based on curve slope analysis are as follows: In 2017, Muhammad calculated the first-order derivative of each pixel in each laser curve in an image coordinate system [12]. If the derivative value exceeded the threshold value, it was determined to be a feature point. However, this method is only appropriate for extracting simple-shaped sheet joint weld feature points, and it is not applicable to the extraction of feature points for cylindrical longitudinal welds. Researchers from Tsinghua University have used the third-order Savitzky–Golay filter to calculate the second-derivative value of the laser fringe curve [13]. If the calculated value exceeded the threshold value, then it was determined as an undercut feature point. In [14], a structured light-vision sensor with a narrowband filter was developed. After the centerline extraction of the sensor output image, the weld width and undercut feature points are extracted by calculating the second-order derivative of the laser stripe centerline. However, the algorithm can only measure the parameter width and undercut. This method is better than the previous curve method, but it is sensitive to noise. The feature point extraction methods based on the corner detection method curve inflection points are as follows. In [15], Kovacevic used a corner detection algorithm to extract the feature points of the weld toe and reinforcement on both sides of the weld of U-shaped, V-shaped, square, and lap joints. However, this method does not consider the feature point extraction of undercut defects. In [16], Hessian eigenvectors were used to locate the position of the weld centerline inflection point as a feature point, where the parameter width and reinforcement can be calculated. Consequently, it cannot be used to measure the undercut and misalignment parameters. Feature point extraction methods, based on curve processing and fitting, include performing a continuous sym8 wavelet transformation on the laser curve in the image coordinate system and using the local maximum value of the sym8 wavelet coefficient as the feature point of the weld undercut [17]. However, a smoothly changing feature point (reinforcement, width) cannot be obtained using this method because it produces sharp protruding points in the curve (undercut defects, etc.). A curve fitting plus derivative method was proposed in [18], which uses the zero point of the second derivative of the fitted curve as the characteristic point of the apex of the V-shaped plate welded joint. In [19], a structured light vision sensor was used to determine feature points

using **NURBS** curve segmentation and fitting. The influence of image noise was significantly reduced via fitting because the derivative of the fitted curve was used as opposed to direct determination of the slope of the curve. However, the fitting curve differed from the curve, which in turn caused feature point position errors. The representative products of active visual weld inspection include the weld measuring instrument used for various groove welds devised by Servo-Robot of Canada [20], which can measure the reinforcement, width, and undercut, and the seam tracking system developed by the British MetaVision Company [21], which can monitor the longitudinal and circumferential seam welding process of welding robots. It is evident that when compared to the passive visual weld inspection method, the active visual weld detection method exhibits a better reflection of the image characteristics of the weld parameters and lower inspection environment requirements. This has become the mainstream method for weld visual tracking, identification, and inspection. However, the applicability of the current feature-point extraction algorithm should be improved. It is not possible for any algorithm to simultaneously detect the four parameters of weld width, reinforcement, undercut depth, and misalignment.

In recent years, deep learning has advanced rapidly in machine vision research. The processing methods for extracting specified feature points from images are classified as pose estimation algorithms. The Deep-Pose network for image feature estimation was first proposed by Google in 2014 [22]. The front end of the network uses deep convolutional neural networks (**CNN**) to extract image feature information at multiple scales, and the back end uses multi-scale features of the convolutional layer. The output is connected to the fully connected layers (**FC**), and the coordinate extraction task of the feature points in the image coordinate system is completed. Owing to the limitation in **CNN**'s feature extraction performance at the time, the Deep-Pose accuracy was low. Megvii Technology (2018) proposed a cascaded pyramid network (**CPN**) [23], which was classified into Global-Net and Refine-Net. Global-Net uses a pyramid structure to extract **CNN** to extract image multiscale features, and Refine-Net structure regresses on image feature points. To accurately estimate the coordinate information of feature points, the complex and inefficient FC layer in Deep-Pose was abandoned, and a simple and efficient regression method of the deconvolution layer and mean pooling layer structure was used. To date, this is the best network for feature point extraction. A simple and effective bottom-up structure of DeepLabCut has been proposed in the literature [24] for pose estimation of animals, and the network exhibits good migration performance and can be applied to feature point extraction of other objects. In [25], a residual step network structure was proposed to fuse features of the same spatial dimension (intra-level features) to further optimize the key point locations. To solve the problem of severe degradation of feature map resolution in the calculation of traditional CNN structures, [26] proposed an HR-net that can maintain a high-resolution feature map output during the convolution calculation. Later, [27] designed a bottom-up human pose estimation structure based on HR-net. Reference [28] proposed a TokenPose network structure, which replaces the image convolution calculation with the transformer structure in the NLP research field. The network feature extraction AP is close to the literature result, but the network structure is lighter, and it is only suitable for human pose estimation. Therefore, this study attempts to combine a deep learning image feature extraction method with active visual weld inspection technology to provide a new research idea for the field of weld inspection.

The main contributions of this study are as follows:

1. We analyzed the reasons due to which the standard welding seam surface parameter measurement method cannot be used for actual pressure pipeline inspections, and we proposed a measurement index for welding seam surface parameters to account for the coexistence of numerous surface parameters.

2. A deep convolutional neural network-based image parameter feature point extraction approach can simultaneously extract all parameter feature points in a single image.

3. We proposed a method based on the third-order NURBS curve for enhancing the image data of the weld surface profile of a pressure vessel, which can significantly reduce the number of deep network data collection tasks.

The remainder of this paper is organized as follows. In the section on modeling and numerical analyses, the design details of a weld surface device based on a laser profile sensor are provided and an overview of the weld surface parameter calculation algorithms are outlined. Various experimental results are provided to confirm the validity of the proposed methods in the results section. Finally, conclusions are provided in the discussion section.

## Modeling and numerical analyses

### Weld surface parameters measuring device based on laser profile sensor

A laser profile sensor, electric slide-in Z-axis, manual slide in Y-axis, and computer are shown in Fig 1 as a system for measuring the weld surface parameters based on a structured light model. A KEYENCE LJ-V7080 laser profile sensor with a 32-mm built-in camera and built-in laser with a wavelength of 405 nm were used in the experiment. The measuring ranges of the sensor in the *X*- and *Y*-axis directions were 20 mm and 46 mm, respectively. The sensor was installed directly above the cylinder for measurement, and its imaging distance was within this range. The one-line-shaped laser emitted by the sensor hit the surface of the pressure vessel to be measured. The

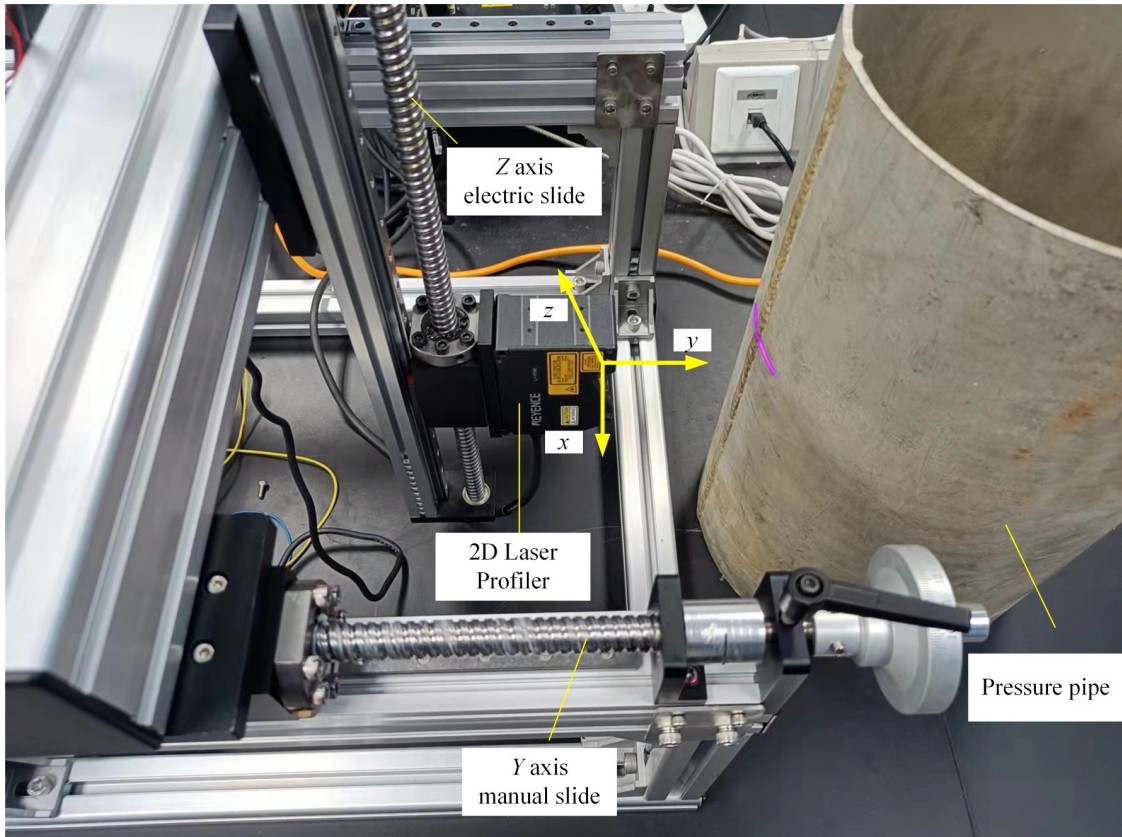

**Fig 1. Device for measuring weld surface parameters, which consists of a laser profile sensor, electric slide-in Z-axis, manual slide in Y-axis, and computer.**

camera inside the sensor captured images near the linear laser range in real time, and the point cloud data were output to the computer via an internal algorithm. The sensor was fixed on the Z-axis electric guide rail, and the line laser emission surface was parallel to the cross-section of the pressure vessel cylinder. The Z-axis electric slide rail was connected to the Y-axis manual slide rail via a bracket, allowing the sensor to move in both directions. The laser sensor and pressure pipe cylinder were kept tangential to the movement direction during the detection procedure.

The laser profile sensor-based welding seam surface profile parameter detection system described in this study is shown in Fig 2. The entire detection process is as follows: ① the laser profile sensor emits a laser that is perpendicular to the weld surface of the pressure vessel; ② point set $S$ generated by the sensor is preprocessed to generate the weld contour curve image $G$; ③ deep learning network, and the characteristic points of the weld parameters are output at coordinates $P(x, y)$; ④ according to the proposed numerical calculation index of the weld surface parameters, the numerical calculation of the parameters is completed.

In general, the weld contour curve image $G$ is generated from the weld contour point set $S$, and this method requires that it can highlight and accurately reflect the surface contour characteristics of the weld. The common image-production method first creates an $m \times n \times 3$ three-channel back matrix $I_{back}$. Then, to generate image $G$, each contour point set was filled with a black background image in the form of monochrome pixels. Although the picture background (black background) and content (contour pixels) scales are extremely small, as low as 1/10000, this image-generating method can precisely depict the surface contour properties of the weld. The model can be over-fitted if the image is used for subsequent deep learning training. Therefore, after the black background image is created using the above method, the following formula is used to generate a contour curve image $G$ with a large ratio between the image background and content. Let $Pix$ be the RGB value of the contour point color and $\langle \cdot \rangle$ be the calculation method of rounding down. Then, $G$ can be expressed as follows:

$$G = \begin{cases} I_{back}(\langle x_{S_i} \rangle, \langle y_{S_i} \rangle, 1) = Pix_R \\ I_{back}(\langle x_{S_i} \rangle, \langle y_{S_i} \rangle, 2) = Pix_G \\ I_{back}(\langle x_{S_i} \rangle, \langle y_{S_i} \rangle, 3) = Pix_B \end{cases} \tag{1}$$

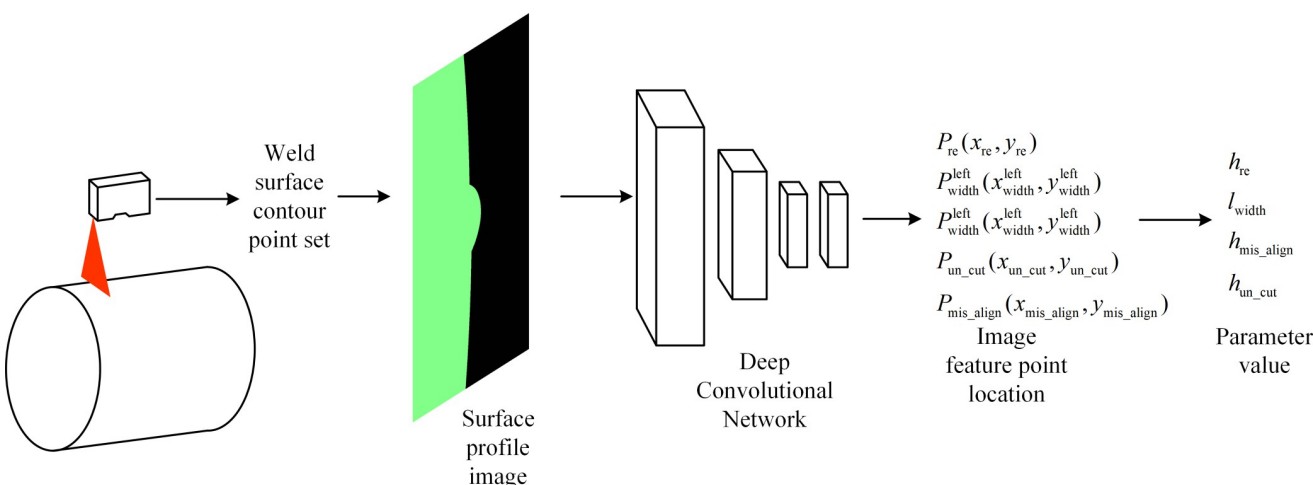

**Fig 2. Flowchart of the laser profile sensor-based welding seam surface profile parameter detection system.**

## Weld surface parameter measurement index with multiple defect parameters

The butt joint longitudinal and girth weld reinforcements, as well as three measurement surface parameters (width, undercut, and misalignment), were defined in accordance with the AWS A3.0 "Definition of Standard Welding Terms" [29], ISO 5817 "Welding Joints" [30], AWS D1.1 "Welding Specification for Steel Structures" [31], and ASME VIII "Boiler and Pressure Vessel Manufacturing Code" standards [32]. A schematic of the four-parameter measurement requirements for butt welds is shown in Fig 3. The parameter corresponding to weld width is defined as the distance between the two weld toes according to the AWS A3.0 "Definition of Standard Welding Terms," which is the junction between the weld surface and base metal. The weld reinforcement $h_{re}$ is a parameter in which the weld metal exceeds the height of the fillet welding groove. The weld parameter undercut $h_{un\_cut}$ denotes the size of grooves or depressions produced along with the base metal of the weld toe owing to improper selection of welding parameters or incorrect operation methods. Fig 3(A) shows the definitions of weld width $l_{width}$, weld reinforcement $h_{re}$, and weld undercut $h_{un\_cut}$ in the standard. Weld misalignment is defined by the ASME VIII "Boiler and Pressure Vessel Manufacturing Code" standard as the phenomenon of dislocation and unevenness due to the deformation of the welding deviation and other factors during welding. The parameter weld misalignment $h_{align}$ denotes the size and amplitude of the misalignment as shown in Fig 3(B).

The existence of weld defects alone, which is a measurement index under the ideal weld condition, is an example of each weld appearance parameter defined and detailed in the standard state. An initially formed weld can exhibit multiple coexisting defects. In the case of weld defects (such as wrong under the influence of edge volume), the definition of the welding seam parameter measurement index in the standard example diagram is no longer applicable. Hence, the appearance parameters of welds at the cross-sectional position of the welds under the conditions of normal welds without defects, single-defect welds, and multi-defect welds are discussed in this study with reference to measurement diagrams of weld appearance parameters in relevant standards. Fig 4 shows a schematic of the weld parameter measurement indicators for various weld shapes.

The parameter reinforcement feature point $P_{re}$ was determined as the highest point of the welding area of the profile curve. The parameter width feature points $P_{width}^{left}$ and $P_{width}^{right}$ are determined as the intersection of the welding curve and the base metal curve, i.e., the welding toes on both sides, as shown in Fig 4(A) and 4(B). The misalignment feature points $P_{align}^{left}$ and $P_{align}^{right}$

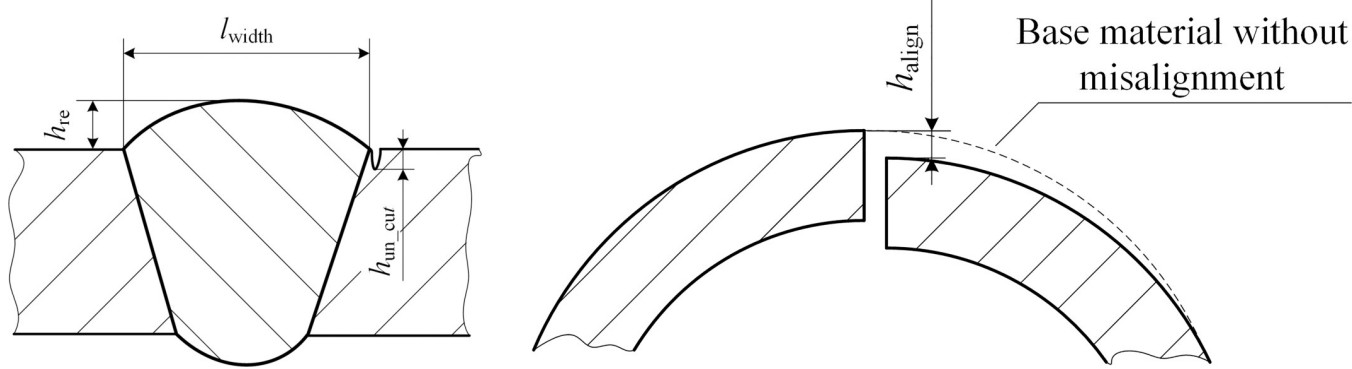

**Fig 3. Schematic diagram of parameter definition of butt weld 4.** (a) Definition of weld reinforcement, width, and undercut. (b) Definition of weld misalignment.

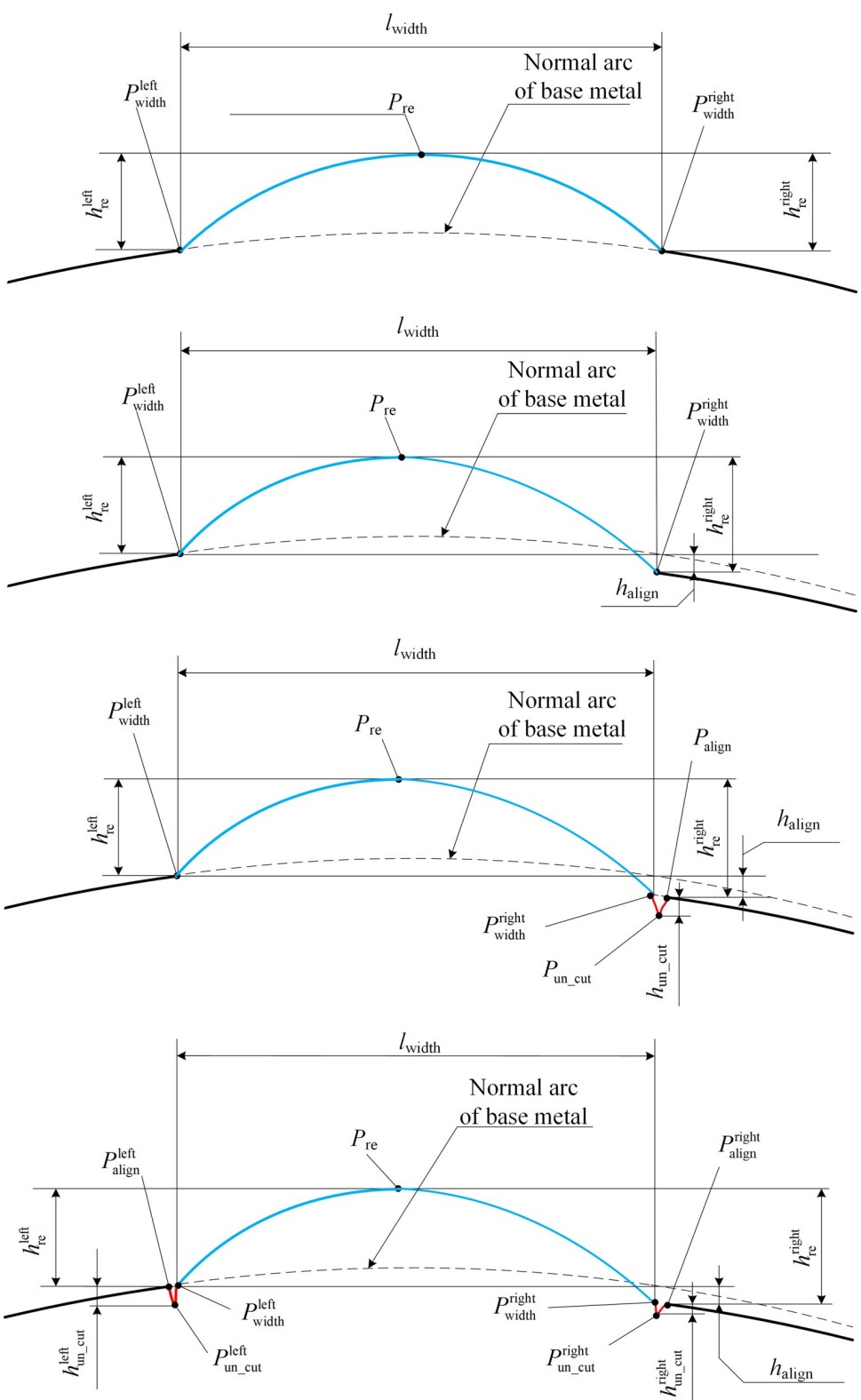

**Fig 4. Schematic diagrams of welding seam parameter measurement index under different welding seam shapes.**
(a) Normal weld. (b) Welds with the wrong sides. (c) Welds with single undercut and misalignment. (d) Welds with double undercut and misalignment.

coincide with the width feature points in the cases of no defect parameters and single misalignment defects. However, given that the weld toe on the undercut side disappears in the case of a defective undercut, the parameter width feature point is modified to the junction point of the undercut depression curve and welding curve. The misalignment feature point at this time is the intersection point between the undercut curve and base material curve as shown in Fig 4 (C) and 4(D), respectively.

The parameter feature points in normal welds are extreme points or corner points, as shown in the above feature point selection example, and the traditional curve extreme and reciprocal analysis methods can complete the feature point extraction task. The characteristic points of the cross-section curve width are weak owing to undercut defects. It is difficult to simultaneously extract all of the characteristic points based on traditional curve analysis methods, and to date, no scholars have proposed a method for simultaneously extracting all of the appearance parameters. Consequently, in terms of image processing feature analysis, in this study, we employed deep learning image semantic segmentation methods to classify weld defects and extracted four parameter feature points of welds in laser curve images.

## Design of image feature point extraction network based on CNN

The structural diagram of the coding-decoding-free image feature point extraction network (**EDE**-net), proposed for this study, is depicted in Fig 5. The input of this network is the pre-processed laser profile image of the weld, and the output of the network is the pixel position of the parameter feature point. The coding part of the network is composed of the **CNN** backbone. First, the **CNN** backbone output feature map outputs an $n$-dimensional feature map $\overline{M}$ after upsampling at the branch of the decoding part. Then, the location of the feature point, roughly extracted by the network, can be expressed as $\left(\overline{x_{\text{rough}}^i}, \overline{y_{\text{rough}}^i}\right) = \text{argmax}\,\overline{M}_i(x, y)$ $(i = 1, \ldots, n)$. On branch two of the decoding part, the output feature map $\overline{N}$ with dimensions of $2n$ is sampled and processed, and feature point correction information $\overline{x_{\text{cor}}^i} = \overline{N_{2i-1}}(x_{rough}^i, y_{rough}^i)\,\overline{y_{\text{cor}}^i} = \overline{N_{2i}}(x_{rough}^i, y_{rough}^i)$ is finely extracted. Finally, the position of the

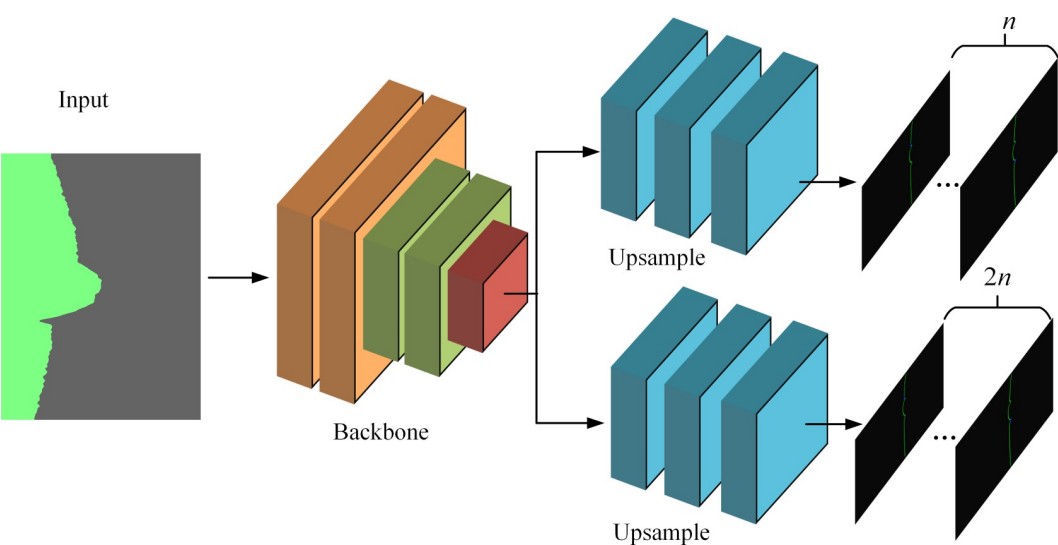

**Fig 5. Network structure diagram of EDE-net.**

parameter feature point in the input image, namely $(\overline{x^i}, \overline{y^i}) = (\overline{x^i_{rough} + x^i_{cor}}, \overline{y^i_{rough} + y^i_{cor}})$, can be obtained by combining the information of the two output feature maps $\overline{M}$ and $\overline{N}$.

**The EDE**-net coding structure output feature map up-sampling methods include bilinear interpolation, deconvolution, and depooling methods. If the image up-sampling multiple is $k_{upsample}$, then the input scale is $ih \times iw \times 3$ image $I$, the up-sampling output scale is $ih^* k_{upsample} \times iw^* k_{upsample} \times 3$ image $O$, and the pixel position $(kp_x, kp_y)$ in the image $O$ pixel value $O_{kp_x, kp_y}$ is sampled and mapped to the pixel position $(p_x, p_y)$ of the image $I$ pixel value $I_{p_x, p_y}$. The up-sampled image $O$ after bilinear interpolation is determined as follows:

$$O_{kp_x, kp_y} = (1-u)(1-v)I_{p_x, p_y} + v(1-u)I_{p_x, p_y+1} + u(1-v)I_{p_x+1, p_y} \tag{2}$$

where $p_x = <\frac{kp_x}{k_{upsample}}> + u, p_y = <\frac{kp_y}{k_{upsample}}> + v$, and $\langle \cdot \rangle$ are round-down calculations. Based on the formula, it is evident that the bilinear interpolation upsampling method should traverse each pixel, which features a large amount of calculation and slower running speed.

The deconvolution upsampling mechanism is shown in Fig 6. The adjacent and surrounding pixels in Image $I$ were interpolated and filled with pixels (usually the pixel value was 0). To obtain the inverse convolution output image $O$, the convolution kernel and supplemented images were used for the convolution calculation. The pooling kernel was set according to the maximum and average pooling methods, and the de-pooling up-sampling mechanism was the same as the deconvolution up-sampling image supplement method. The up-sampling magnification of the deconvolution and de-pooling methods was lower, but the amount of calculation was less. In deconvolution, the convolution kernel can participate in the entire network training and update the weights, whereas the weights of the pooling kernel cannot be changed. Therefore, **EDE**-net uses the deconvolution calculation as the up-sampling method.

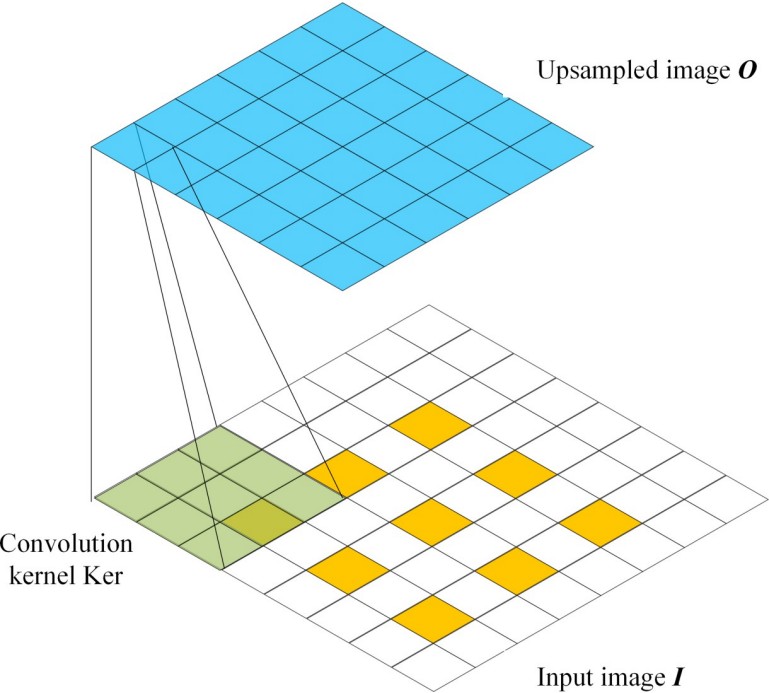

**Fig 6. Deconvolution up-sampling mechanism diagram.**

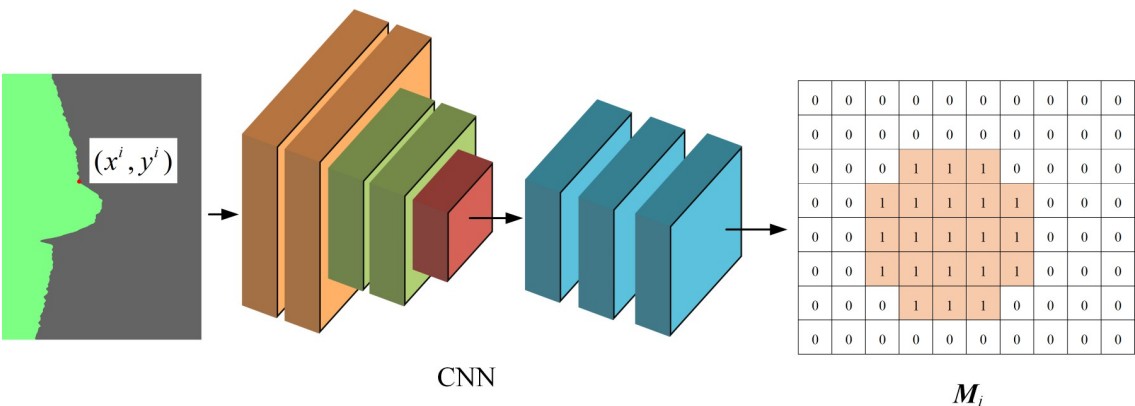

**Fig 7. Theoretical output results of EDE-net branch 1.**

**The EDE**-net network branch is the task of rough extraction of feature point positions. Based on the concept of completely convolutional network semantic segmentation, the deep feature map output by the convolutional network was processed via a single layer with a stride of 2 and a 3 × 3 deconvolution kernel with a number of feature points *n*. The output dimension of the feature map was reduced to *n*, with each dimension corresponding to the approximate position information of each feature point in the input weld centerline image.

Each pixel value $\overline{M}$ is the probability that the feature point is at that position. The theoretical output result, M, of the feature extraction module branch is shown in Fig 7. If the theoretical regression object $M$ is the only single-point position of the characteristic point in the laser centerline image of the weld, then the position outside the characteristic point in $M$ is a negative sample. If the positive and negative samples are seriously out of balance, overfitting can occur during network training. Therefore, the feature point distance threshold T was introduced, and the pixel distance from the feature point position was set as positive samples within *T*. This in turn solved the imbalance problem. The theoretical output feature map $M$ can be obtained as follows:

$$M_{m_x,m_y,m_i} = \begin{cases} 1, (m_x - x^i)^2 + (m_y - y^i)^2 \leq T \\ 0, \text{else} \end{cases}, i = 1, \cdots, n \qquad (3)$$

The distances between the feature point and background within the threshold $T$ can be treated as a binary classification task. Hence, the focal loss of the feature classification task can be utilized as the loss function as shown in the output feature map of the branch-network theory. The adjustment parameters $\alpha$ and $\gamma$ are compatible with the feature classification task Focal-loss [33]. Therefore, the branch-loss function $L_M$ is as follows:

$$L_M = \sum_{i}^{n} -\alpha(S - PT)^\gamma \log PT, \ PT = M_{m_i} * \overline{M}_{m_i} + (S - M_{m_i}) * (S - \overline{M}_{m_i}) \qquad (4)$$

where matrix $S$ elements are all one. **EDE**-net branch 2 is a feature point position–correction task. The convolutional network's deep feature map is processed through a single layer with a stride of 2, scale of 3×3, and dimension of 2n deconvolution kernels. This yields a 2n-dimensional feature map $\overline{N}$. The theoretical output results $N$ of **EDE**-net network branch 2 are shown in Fig 8. Based on the approximate location of $M$ feature points, the branch two-theory output feature map N adds a feature point location correction value. This implies that the

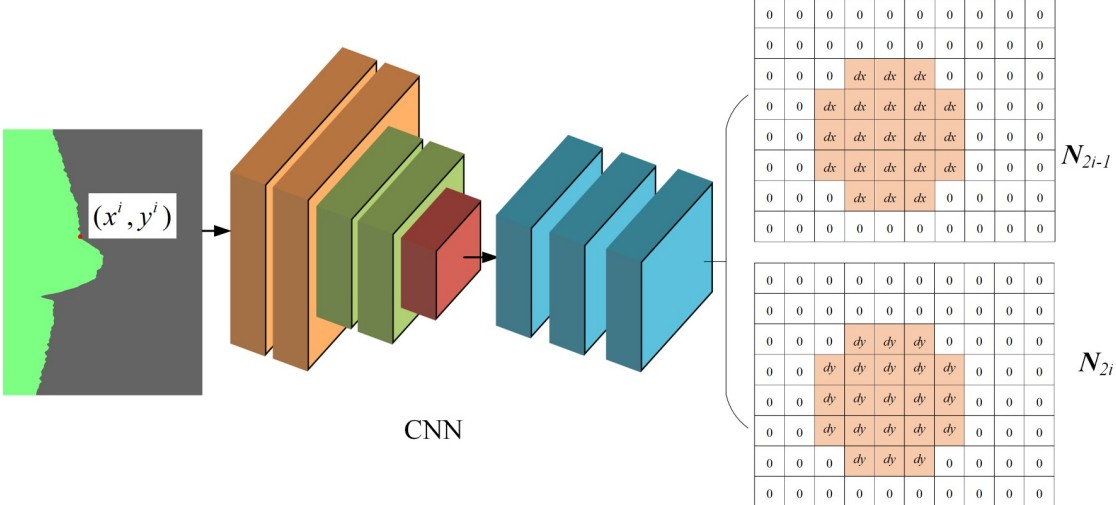

**Fig 8. Theoretical output results of EDE-net branch 2.**

position value of the corresponding element of $N_{2i-1}$ is the true value of $M_i$, and $N_{2i}$ is the pixel difference between the theoretical feature points and *X-Y-axis* of the image coordinate system at the element's position. Therefore, the theoretical outputs $N_{2i-1}$ and $N_{2i}$ branch 2 are defined as follows:

$$N_{n_x,n_y,n_{2i-1}} = \begin{cases} c(n_x - x^i), (n_x - x^i)^2 + (n_y - y^i)^2 \leq T \\ 0, \text{else} \end{cases}$$

$$N_{n_x,n_y,n_{2i-1}} = \begin{cases} c(n_y - y^i), (n_x - x^i)^2 + (n_y - y^i)^2 \leq T \\ 0, \text{else} \end{cases} \quad (5)$$

where $c$ denotes the ratio of the input image scale to the output feature map scale of upsampling. The feature point position correction task is a numerical regression task. Therefore, the Huber loss is used to establish the loss function:

$$L_N = \sum_i^n l_i, l_i = \begin{cases} c(\sum_{n_x,n_y} |N - \overline{N}|)^2, \sum_{n_x,n_y} |N - \overline{N}| < 1 \\ \sum_{n_x,n_y} |N - \overline{N}| - c \sum_{n_x,n_y} |N - \overline{N}|^2, \text{else} \end{cases} \quad (6)$$

Then, the **EDE**-net network loss function can be obtained as $L_{EDE} = L_N + L_M$.

## Image data enhancement method for pressure vessel weld surface profile based on third-order NURBS curve

A **CNN** requires a large number of datasets for training, and the data capacity of the training set is directly related to the **CNN**'s ability to extract feature points. The conventional approach of producing training sets involves acquiring contour images with an active vision imaging device and manually labeling the location coordinates of the feature points in each image. To avoid this, in the current study, we provide a surface parameter simulation method with the

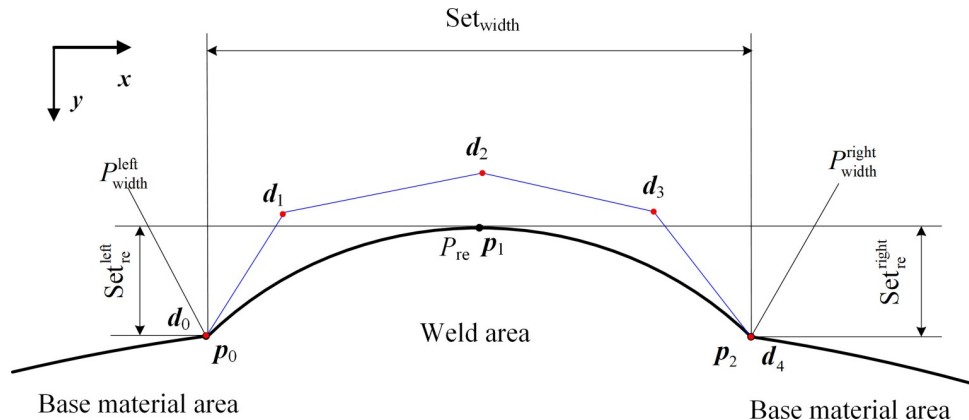

**Fig 9. Simulation of normal weld curve.**

coexistence of multiple defects on the weld surface of the pressure vessel. This in turn allows variation of the types of weld parameters, number of feature points, and parameter values, and thereby, effectively reduces the number of training set collection tasks.

A simulation diagram of the normal weld curve is shown in Fig 9. The parent material area of the curve in the image coordinate system ($F_{\mathrm{metal}}(x,y)$) is expressed as follows:

$$F_{\mathrm{metal}}(x, y) = \begin{cases} (x - L_{\mathrm{pic}}/2)^2 + (y - W_{\mathrm{pic}}/2 - R_{\mathrm{stand}})^2 = R_{\mathrm{stand}}^2 \\ x \in [0, L_{\mathrm{pic}} - \mathrm{Set}_{\mathrm{width}}] \cap [(L_{\mathrm{pic}} + \mathrm{Set}_{\mathrm{width}}), W_{\mathrm{pic}}] \end{cases} \tag{7}$$

where $\mathrm{Set}_{\mathrm{width}}$ denotes the weld width parameter, $R_{\mathrm{stand}}$ denotes the diameter of the pressure vessel cylinder, $L_{\mathrm{pic}}$ denotes the imaging area length, and $W_{\mathrm{pic}}$ denotes the width.

The simulation feature point position of the normal weld contour curve is presented in Table 1 when the simulation reinforcement is $\mathrm{Set}_{\mathrm{re}}$.

A non-uniform rational B-spline (**NURBS**) curve [34] simulation weld zone curve was constructed based on the aforementioned three points: $P_{\mathrm{re}}$, $P_{\mathrm{width}}^{\mathrm{left}}$, and $P_{\mathrm{width}}^{\mathrm{right}}$. Assuming that the control point of the third-order **NURBS** curve is $k = 3$, $d_i = [x, y]^T$, $w_i$ is the weight factor of the curve control point, and $w_0$, $w_2 > 0$, the remaining $w_i \geq 0$, $F_{\mathrm{weld}}(u)$ can be expressed as follows:

$$F_{\mathrm{weld}}(u) \frac{\sum_{i=0}^{n} w_i d_i N_{i,k}(u)}{\sum_{i=0}^{n} w_i N_{i,k}(u)}, u \in [0, 1] \tag{8}$$

$$N_{i,0}(u) = \begin{cases} 1, u_i \leq u \leq u_{i+1} \\ 0, else \end{cases}, N_{i,k}(u) = \frac{u - u_i}{u_{i+k} - u_i} N_{j,k-1}(u) + \frac{u_{i+k+1} - u}{u_{i+k+1} - u_{i+1}} N_{i+1,k-1}(u) \tag{9}$$

**Table 1. Simulation feature point position of normal weld contour curve.**

| Feature point type | Image coordinate system $X$ coordinate | Image coordinate system $Y$ coordinate |
|---|---|---|
| $P_{\mathrm{re}}$ | $L_{\mathrm{pic}}$ | $F_{\mathrm{metal}}(x, y)\big|_{L_{\mathrm{pic}} - \mathrm{Set}_{\mathrm{width}}} + \mathrm{Set}_{\mathrm{width}}$ |
| $P_{\mathrm{width}}^{\mathrm{left}}$ | $L_{\mathrm{pic}} - \mathrm{Set}_{\mathrm{width}}$ | $F_{\mathrm{metal}}(L_{\mathrm{pic}} - \mathrm{Set}_{\mathrm{width}})$ |
| $P_{\mathrm{width}}^{\mathrm{right}}$ | $L_{\mathrm{pic}} + \mathrm{Set}_{\mathrm{width}}$ | $F_{\mathrm{metal}}(L_{\mathrm{pic}} + \mathrm{Set}_{\mathrm{width}})$ |

Furthermore, $d_i$ information can be inversely calculated with three **NURBS** curve data points: $P_{re}$, $P_{width}^{left}$, and $P_{width}^{right}$. Let the node vectorbe $U = [u_0, u_1, u_2, \cdots, u_{n+4}]$ when the curve is opened and the control point be

$n = m+k-1 = 5$ such that the curve passes through the first and last control points. The node vector should exhibit $k+1$ repeatability, and it is set using the accumulation chord length method. This implies that $u_0 = u_1 + u_2 = u_3 = 0$, $u_5 = u_6 + u_7 = u_8 = 1$,

$u_4 = |\boldsymbol{p}_1 - \boldsymbol{p}_0|/\sum\limits_{i=1}^{n} |\boldsymbol{p}_i - \boldsymbol{p}_{i-1}|$, and the setting of the weight factor $w_i$ of the curve control point are as follows:

$$\sum_{i=0}^{n+2} w_i N_{i,k}(u_{j+k}) = 1, j = 0, 1, \cdots, n \tag{10}$$

The beginning and end points of the curve control points were consistent with those of the data points throughout the reverse solution procedure. The connection points of the curved segment correspond to the nodes of the **NURBS** curve-defining domain. Therefore, if the data point $q_i$ corresponds to the node value $u_{i+3}(i = 0,1,2)$, then the solution condition for $d_i$ is as follows:

$$\frac{\sum\limits_{j=i-k}^{i} w_j \boldsymbol{d}_j N_{j,k}(u_i)}{\sum\limits_{j=i-k}^{i} w_j N_{j,k}(u_i)} = \boldsymbol{q}_{i-3}, u \in [u_i, u_{i+1}] \subset [u_3, u_5], i = 3, 4, 5 \tag{11}$$

To complete the control-point solution provided by the tangent vector boundary conditions, the following two equations must be included:

$$F'_{weld}(0) = \frac{kw_1|\boldsymbol{d}_1 - \boldsymbol{d}_0|}{w_0(u_{n-1} - u_{n-2})}, \ F'_{weld}(1) = \frac{kw_{n-2}|\boldsymbol{d}_{n-1} - \boldsymbol{d}_{n-2}|}{w_{n-1}(u_n - u_{n-1})} \tag{12}$$

By combining the above equations, we can complete the coordinate position solution $\boldsymbol{d}_i$ and construct curve $F_{weld}(u)$.

**Defect parameter undercut simulation.** If the undercut width, depth, and offset are $\delta_{un\_cut}$, $Set_{un\_cut}$, and $\Delta_{un\_cut}$, respectively, then they indicate that the undercut feature point and width feature point are separated in the $X$-axis direction. Then, the value range of $x$ in the curve of $F_{metal}^{right}(x, y)$ of the parent metal area on the right is modified to $x \in [L_{pic} + Set_{width} - \delta_{un\_cut}, W_{pic}]$. The positions of the simulated feature points of the undercut weld profile curve with defect parameters are listed in Table 2.

To perform the single-defect undercut weld contour curve simulation, five-point coordinates $P_{re}$, $P_{width}^{left}$, $P_{width}^{right}$, $P_{un\_cut}$, and $P_{mis\_align}$ can be used as the **NURBS** curve data points as shown in Fig 10.

**Defect parameter misalignment simulation.** In this case, the width feature points on both sides of the weld coincide with the deviation feature point, and curve $F_{metal}^{left}(x, y)$ of the base material area on the left is consistent with the simulation of the normal weld. The curve of the base material area on the right $F_{metal}^{right}(x, y)$ by $Set_{mis\_align}$ along the $Y$-axis of the image

**Table 2. Location of the feature point in the simulation of the undercut weld profile curve with the defect parameter.**

| Feature point type | Image coordinate system $X$ coordinate | Image coordinate system $Y$ coordinate |
|---|---|---|
| $P_{re}$ | $L_{pic}$ | $F_{metal}(x, y)\big|_{L_{pic}-Set_{width}} + Set_{width}$ |
| $P_{width}^{left}$ | $L_{pic}-Set_{width}$ | $F_{metal}(L_{pic}-Set_{width})$ |
| $P_{width}^{right}$ | $L_{pic}+Set_{width}$ | $F_{metal}(L_{pic}+Set_{width})$ |
| $P_{un\_cut}$ | $L_{pic}+Set_{width}-\delta_{un\_cut}-\Delta_{un\_cut}$ | $F_{metal}^{right}(L_{pic}+Set_{width}-\delta_{un\_cut})-Set_{un\_cut}$ |
| $P_{mis\_align}$ | $L_{pic}+Set_{width}-\delta_{un\_cut}$ | $F_{metal}^{right}(L_{pic}+Set_{width}-\delta_{un\_cut})$ |

coordinate system and $F_{metal}^{right}(x, y)$ on the right side of the weld is as follows:

$$F_{metal}^{right}(x, y) = \begin{cases} (x - L_{pic}/2)^2 + (y - W_{pic}/2 - R_{stand} - Set_{mis\_align})^2 = R_{stand}^2 \\ x \in [L_{pic}+Set_{width}, W_{pic}] \end{cases} \tag{13}$$

The single-defect misaligned weld profile curve simulation was performed using the coordinates $P_{re}$, $P_{width}^{left}$, and $P_{width}^{right}$. The three points were used as **NURBS** curve data points as shown in Fig 11.

## Results

### Experiment on EDE-net performance of different backbone networks

**Test CNN backbone selection.** The accuracy of the overall network feature point extraction is affected by the feature extraction performance of the CNN backbone network in the image feature point extraction network based on encoding-decoding. Among the common CNN networks, including AlexNet [35], VGG [36], Res-Net [37], and Inception [38], AlexNet and VGG networks are linear and branchless structures. When the network layers are deeper, they are more difficult to develop, and common problems, such as gradient disappearance and explosion, can occur. To solve the aforementioned problems, Res-Net introduces the residual unit residual in the convolutional layer and realizes identity mapping by constructing direct connections. The network does not degrade as the depth of the network convolutional layer increases owing to the continued stacking. The inception network proposed a structure to obtain close feature extraction capabilities with fewer network layers. The feature map was produced using several convolutions and pooling kernels, and the results were stacked to

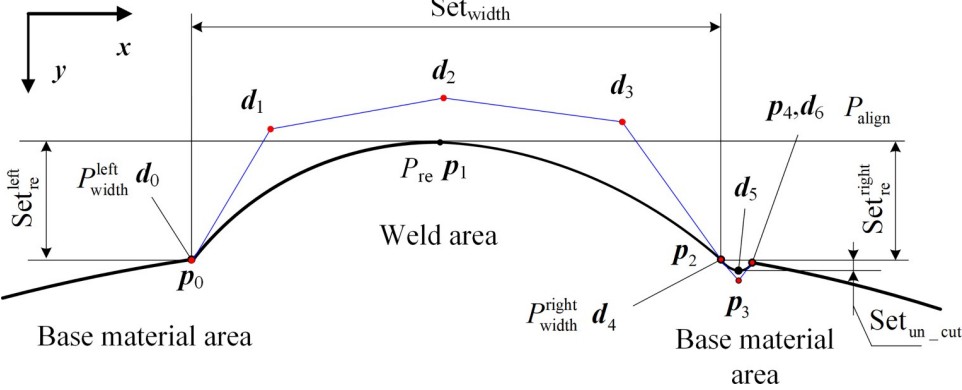

**Fig 10. Simulation of weld profile curve with single defect parameter undercut $Set_{un\_cut}$.**

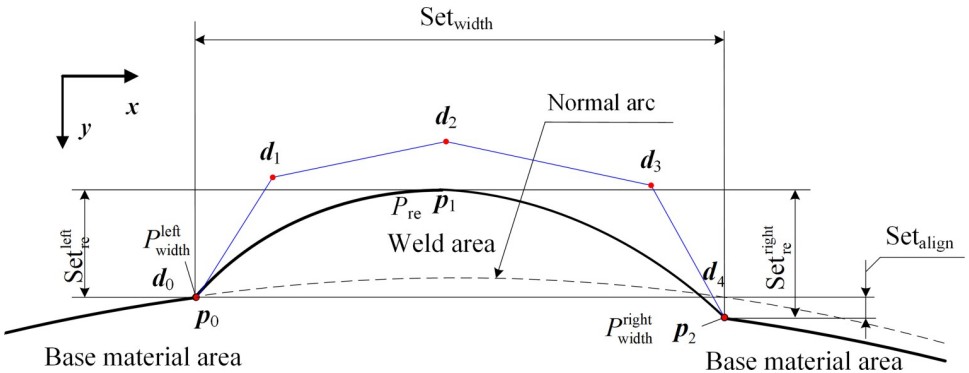

**Fig 11. Simulation of weld profile curve with single defect parameter misalignment $Set_{align}$.**

reduce the number of layers in the network. Resnet50, Resnet101, Resnet152, InceptionV4, and Inception-Res-net were used as the backbones of the CNN based on the encoding-decoding deep feature-point extraction network. Table 3 lists the tested CNN network information, where Top1 accuracy is the CNN structure in the image net image classification result. This can be used as the performance level of the network. The trainable parameters indicate the complexity of the network training owing to the involvement of many parameters, such as the standard of performance, higher accuracy, and better network performance.

**Selection of training sets.** Based on the existence of the undercut defect parameter, the number of feature points of the weld parameters in images are 3, 5, and 7, namely, datasets $D_3$, $D_5$, and $D_7$. To collect boiler butt type B and pressure pipeline type A welds with diameters of 1300 and 255 mm, respectively, a Keyence LJV-7080 sensor was used. The set of surface contour points comprised the training and test sets. Fig 12 shows the simulation-generated contour image effect and previously acquired contour maps.

To improve the data from the aforementioned photographs, an affine transformation was adopted. Let the image size be $W_D \times H_D$, image rotation angle be $\beta$, and image scaling size be $h_{scale}$. Then, the affine matrix is determined as $\boldsymbol{W}_{warp}$ as follows:

$$
\boldsymbol{W}_{warp} = \begin{bmatrix} 1 & 0 & -W_D/2 \\ 0 & 1 & -H_D/2 \\ 0 & 0 & 1 \end{bmatrix} \begin{bmatrix} \cos\beta & -\sin\beta & 0 \\ \sin\beta & \cos\beta & 0 \\ 0 & 0 & 1 \end{bmatrix} \begin{bmatrix} 1 & 0 & W_D/2 \\ 0 & 1 & H_D/2 \\ 0 & 0 & 1 \end{bmatrix}
$$

$$
\times \begin{bmatrix} W_D/h_{scale} & 0 & -W_D{}^2 h_{scale}/2 + 0.5 \\ 0 & H_D/h_{scale} & -H_D{}^2 h_{scale}/2 + 0.5 \\ 0 & 0 & 1 \end{bmatrix} \tag{14}
$$

**Table 3. Tested CNN network information.**

| CNNs | Top1 accuracy | Trainable parameters | GFLOPs |
|---|---|---|---|
| Resnet50 | 75.2 | 23,508,032 | 11.7 |
| Resnet101 | 76.4 | 42,500,160 | 19.2 |
| Resnet152 | 76.8 | 58,143,808 | 26.7 |
| InceptionV4 | 80.2 | 41,111,232 | 247.5 |
| Inception Res-net | 80.4 | 26,855,264 | 269.9 |

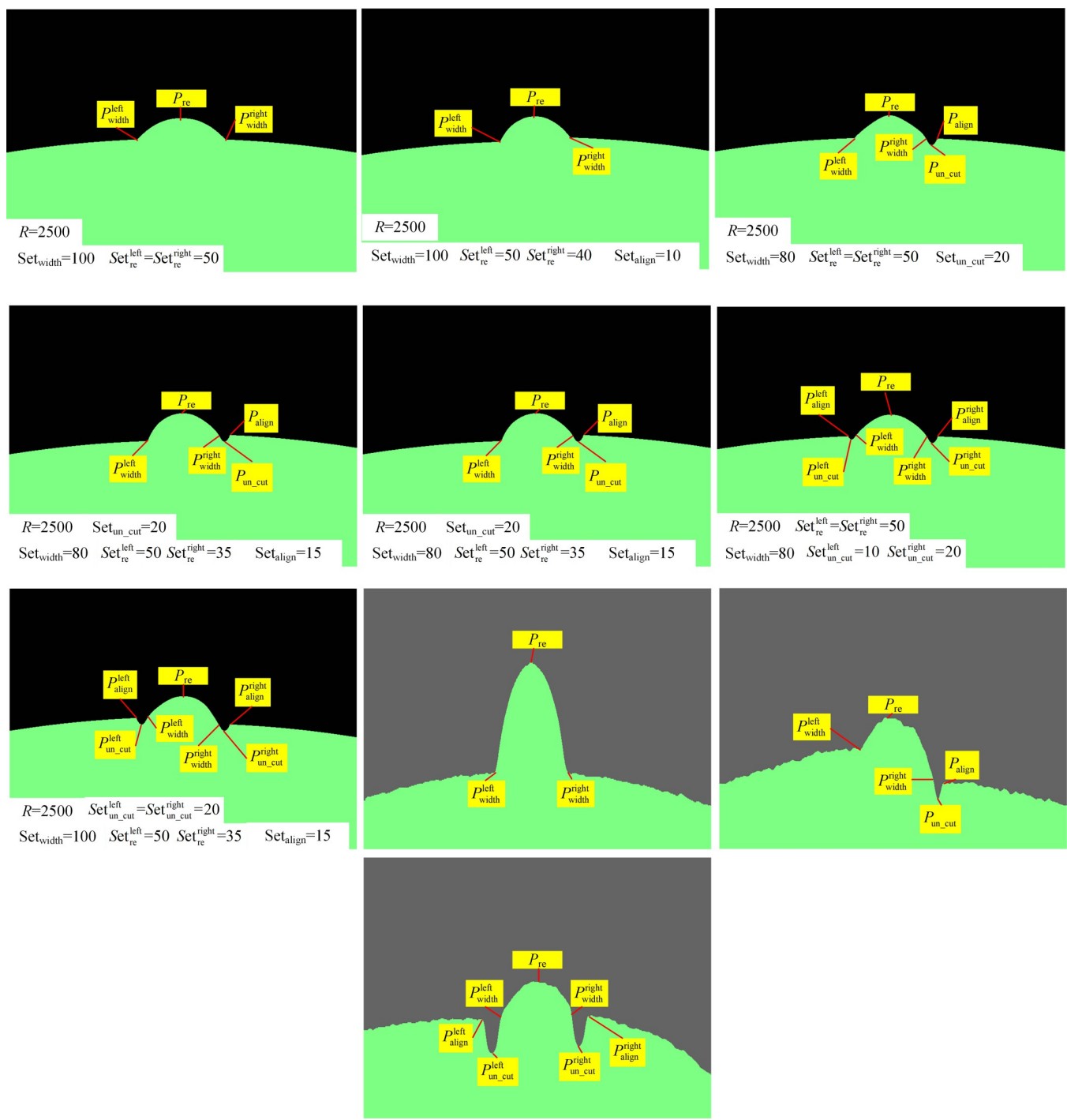

**Fig 12. Acquisition of contour images to conduct simulations to generate contour images.** (a) No defect parameter welds profile data set $D_3$. (b) Weld profile data set $D_3$ for parameter simulation of defects with a single misalignment error. (c) Weld profile data set $D_5$ for parameter simulation of single undercut defect. (d) Weld profile data set $D_5$ for parameter simulation of undercut and misalignment defect parameters. (The undercut is on the misalignment side.) (e) Weld profile data set $D_5$ for parameter simulation of undercut and misalignment defect parameters. (The undercut is not on the misalignment side.) (f) Weld profile data set $D_7$ with parameter simulation of undercut defects on both sides. (g) Weld profile data set $D_7$ for simulation of the misalignment and undercut parameters on both sides (h) Actual collection data set $D_3$. (j) Actual collection data set $D_5$. (j) Actual collection data set $D_7$.

**Table 4. Training hyper-parameter information.**

| Steps | Learning rate |
|---|---|
| 10000 | 0.005 |
| 43000 | 0.02 |
| 73000 | 0.002 |
| 103000 | 0.001 |

One hundred physical collection datasets $D_3$, $D_5$, $D_7$, and 100 simulation-generated datasets $D_3$, $D_5$, and $D_7$ were selected, with a random rotation angle $\beta = 0–30°$ and random scaling size $h_{scale} = 0.5\sim0.8$. Table 4 contains the training of hyperparameter information, and the network training loss function is used as the evaluation index to determine the ideal **CNN** structure.

Fig 13(A)–13(C) show the trend charts of the network loss function extracted from different CNN structures based on the encoding-decoding depth feature points. In the figure, it can be observed that (a) the **CNN** network structure is more difficult to train without fine-tuning migration, resulting loss does not converge, and network model fails. As the number of training steps increased, the training difficulty of **the CNN** structure after fine-tuning dramatically decreased, and the ultimate convergence effect improved. (b) Network training is more effective when the layers of the same **CNN** structure are much deeper. (c) As the performance improves, the accuracy rate of **the CNN** structure increases. For instance, the Inception-Resnet network structure exhibited a Top1 accuracy of 80.4. The training loss function is reduced when compared to that of the ResNet series **CNN**.

## Experiment on the actual pressure vessel weld of measurement results

The experiment was initiated by selecting 150 physical collection datasets and 60 simulation datasets $D_3$, $D_5$, and $D_7$ with a random rotation angle $\beta = 0–30°$ and random scaling size $h_{scale} = 0.5–0.8$. Among the datasets, 90 physical collection datasets and 60 physical collection training sets with 60 simulation-generated training sets were used to train DeepLabCut, HR-net, and **EDE**-net (Inception Res-net), respectively. Finally, the remaining 60 physical collection datasets $D_3$, $D_5$, and $D_7$ were used as the test set, and the accuracy of the parameter feature point extraction was used as the network performance evaluation index.

All the parameter feature points were regarded as feature points of the same nature if the Euclidean distance between the artificially marked parameter feature point and network output feature point was used as the evaluation index. The parameter reinforcement and undercut

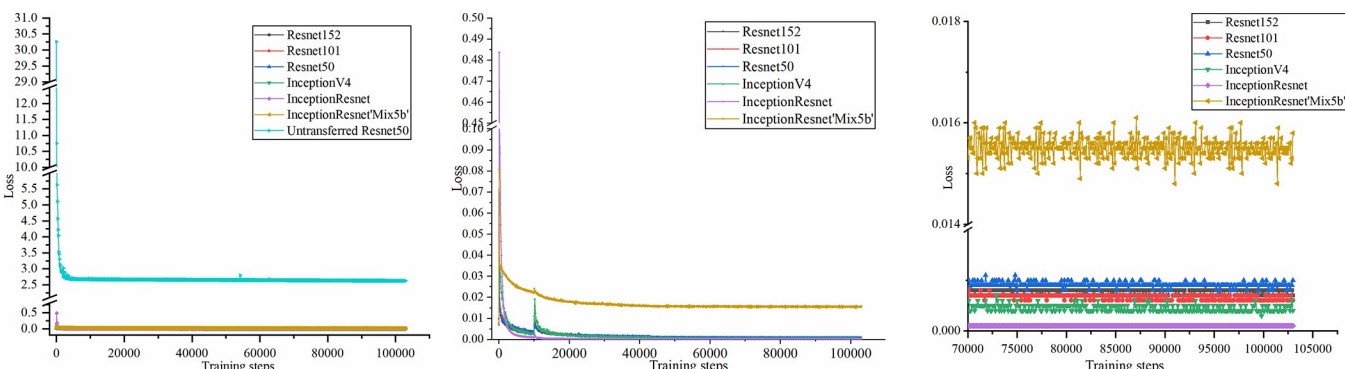

**Fig 13. Loss function trend chart of different CNN backbones based on the encoding–decoding depth feature point extraction network.** (a) Comparison of transfer learning and non-transfer learning. (b) Comparison of loss functions of different network backbones. (c) Final convergence of the network.

feature points are extreme-value natural points. The characteristic reflection is stronger when compared to the parameter width and natural point of the wrong edge inflection point. Given that the feature points of the weld parameters differ in marking and extraction difficulty, using the Euclidean distance as the evaluation index is no longer appropriate. Consequently, the object key-point similarity (OKS) weighted Euclidean distance was introduced as the characteristic evaluation index of the network output parameters. It is defined as follows:

$$OKS_{\bar{S}} = \frac{\sum_i \exp(-d_{\bar{S}}^i/2A_{\bar{S}}^i\sigma_i^2)}{\sum_i 1} \qquad (15)$$

where $\bar{S}$ denotes the presence of an undercut in the weld image, $d_{\bar{S}}^i$ denotes the square of the Euclidean distance between the artificially marked parameter feature point and network output feature point, $A_{\bar{S}}^i$ denotes the pixel area of the weld curve in the image, $\sigma_i^2$ denotes the artificially marked feature point and real position deviation information of the feature point, and $d_{\bar{S}}^i/A_{\bar{S}}^i$ is replaced by the mean square error in the numerical calculation.

Currently, average precision (AP) is used to evaluate the same task in a deep network and performance evaluation index of different network structures. The feature points of the OKS and AP indicators contain AP evaluation methods. The correlation between the tasks is as follows:

$$AP@T_s = \frac{\sum_{\bar{S}}\delta(OKS_{\bar{S}} > T_s)}{\sum_{\bar{S}} 1} \qquad (16)$$

where $T_s$ denotes the OKS threshold. The network output feature points deviate from the actual feature points. The accuracy of the feature point extraction can be improved further if the network output feature points are returned to the laser curve using the following equation.

$$(x_{Cor}, y_{Cor}) = \text{argmin}((x_{out} - x_R)^2 + (y_{out} - y_R)^2)$$

where $(x_{out} - y_{out})$ denotes the coordinate of the characteristic point of the welded seam output by the network, $(x_{Cor}, y_{Cor})$ denotes the coordinate of the characteristic point of welding on the laser line after correction, and $(x_R - y_R)$ denotes the coordinate of any point on the contour line. The network results are presented in Table 5.

**Table 5. AP results of feature points extracted from different CNNs.**

| Methods | CNNs | Training sets | AP$_{0.5}$ | AP$_{0.7}$ | mAP |
|---|---|---|---|---|---|
| Before correction | Deep Lab-Cut | 90 | 0.68 | 0.46 | 0.42 |
| | | 60+60 | 0.63 | 0.55 | 0.44 |
| | HR-net | 90 | 0.42 | 0.29 | 0.24 |
| | | 60+60 | 0.59 | 0.43 | 0.33 |
| | **EDE**-net (Inception Res-net) | 90 | 0.51 | 0.51 | 0.45 |
| | | 60+60 | 0.65 | 0.55 | 0.42 |
| After correction | Deep Lab-Cut | 90 | 0.80 | 0.64 | 0.55 |
| | | 60+60 | 0.79 | 0.65 | 0.49 |
| | HR-net | 90 | 0.81 | 0.68 | 0.57 |
| | | 60+60 | 0.79 | 0.68 | 0.55 |
| | **EDE**-net (Inception Res-net) | 90 | 0.86 | 0.78 | 0.62 |
| | | 60+60 | 0.83 | 0.75 | 0.64 |

**Table 6. Feature point extraction error information.**

| Feature point type | Width feature points | Reinforcement feature points | Undercut feature points | Misalignment feature points |
|---|---|---|---|---|
| Absolute error of Dx | -0.963 | -0.250 | 0.394 | 1.217 |
| Standard deviation of Dx | 1.774 | 1.962 | 0.396 | 2.264 |
| Absolute error of Dy | 0.881 | -0.115 | 0.612 | -0.068 |
| Standard deviation of Dy | 1.426 | 1.413 | 0.965 | 2.224 |

As shown in Table 5, **EDE**-net (Inception-Resnet) before and after correction of AP, AP0.5, AP07 is better than Deep Lab-Cut,and HR-net network, and these corrections can significantly improve the feature point. The extraction accuracies were similar for the two training sets. However, the training method for actual measurements and simulations can effectively reduce the data-collection workload.

Table 6 lists the absolute error and standard deviation data for all the parameter feature points. The measurement resolutions of the X-axis and Y-axis of the sensor were 0.005 and 0.001 mm, respectively. If we select a 99.73% confidence interval [$\mu$-3$\sigma$, $\mu$+3$\sigma$], then the confidence interval of the reinforcement feature point extraction error is [-10.958, 10.8629], and the theoretical measurement accuracy can be as high as 0.011 mm. Similarly, for other parameters, in which the measurement accuracies are mentioned, the confidence interval of the width feature point extraction error was [-9.182, 13.069], and the theoretical measurement accuracy was as high as 0.065 mm. The confidence interval of the extraction error of the undercut feature points was [-10.245, 8.885], and the theoretical measurement accuracy was as high as 0.011 mm. The confidence interval of the error of the feature point extraction of the amount of error was [-11.833, 11.833], and the theoretical measurement accuracy was as high as 0.012 mm.

## Discussion

The measurement indicators for the surface parameters of a single weld specified in the relevant verification standards for pressure vessels cannot be effectively used in the measurement of an actual weld surface profile where multiple defects coexist. In this study, the appearance characteristics of the weld surface parameters were measured in the form of image feature points, and algorithm design ideas of the regression image from the feature point coordinates were proposed using the excellent nonlinear mapping ability of **CNN** networks. An image feature point extraction network based on deep learning was designed to simultaneously extract all the parameter feature points. For the network training measurement, a method based on the 3rd **NURBS** curve simulation of a realistic weld surface profile was proposed to enhance the training data. Finally, an experimental device was designed to collect the surface data of A and B butt welds, and the deep learning network proposed in this study was compared with the DeepLabCut and HR-net methods under different training sets. The results show that the difference between the training output of the training set network after data enhancement and training set network output AP, which is completely measured, is low. However, the data enhancement method can effectively reduce the workload of sample collection, and the theoretical accuracy of parameter measurement can be realized within 0.065 mm.

## Author Contributions

**Data curation:** Pu Liao.

**Formal analysis:** Pu Liao, Liu Guixiong.

**Methodology:** Liu Guixiong.

**Writing – original draft:** Pu Liao, Liu Guixiong.

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
