## [Decision Letter · Decision Letter 0]

16 Feb 2022

PONE-D-22-00457Pressure vessel-oriented visual inspection method based on deep learningPLOS ONE

Dear Dr. Guixiong,

Thank you for submitting your manuscript to PLOS ONE. After careful consideration, we feel that it has merit but does not fully meet PLOS ONE’s publication criteria as it currently stands. Therefore, we invite you to submit a revised version of the manuscript that addresses the points raised during the review process.

We look forward to receiving your revised manuscript.

Kind regards,

Antonio Riveiro Rodríguez, PhD

Academic Editor

PLOS ONE

Journal Requirements:

● A clean copy of the edited manuscript (uploaded as the new *manuscript* file).

6. We note that you have indicated that data from this study are available upon request. PLOS only allows data to be available upon request if there are legal or ethical restrictions on sharing data publicly. For more information on unacceptable data access restrictions, please see http://journals.plos.org/plosone/s/data-availability#loc-unacceptable-data-access-restrictions. 

Reviewers' comments:

Reviewer's Responses to Questions

**Comments to the Author**

1. Is the manuscript technically sound, and do the data support the conclusions?

Reviewer #1: Yes

2. Has the statistical analysis been performed appropriately and rigorously? 

Reviewer #1: Yes

3. Have the authors made all data underlying the findings in their manuscript fully available?

Reviewer #1: Yes

4. Is the manuscript presented in an intelligible fashion and written in standard English?

Reviewer #1: Yes

5. Review Comments to the Author

Reviewer #1: This study presented by the authors is well organized, the presentation of the article and the results are close to satisfactory. However, the reviewed literature studies are not up-to-date, the article should be reupload by examining the recent studies (especially 2020- 2021).

6. PLOS authors have the option to publish the peer review history of their article (what does this mean?). If published, this will include your full peer review and any attached files.

Reviewer #1: No

---

## [Author Response · Author response to Decision Letter 0]

27 Mar 2022

Respones to Editor and Reviewer

Dear Editor and Reviewer:

Thank you for your letter and for the reviewers’ comments concerning our manuscript entitled “Pressure vessel-oriented visual inspection method based on deep learning” (ID: PONE-D-22-00457). Those comments are all valuable and very helpful for revising and improving our paper, as well as the important guiding significance to our researches. We have studied comments carefully and have made correction which we hope meet with approval. Revised portion are marked with “Track Changes” in the paper. The main corrections in the paper and the responds to the reviewer’s comments are as flowing:

Journal Requirements:

1: Please ensure that your manuscript meets PLOS ONE's style requirements, including those for file naming.

Response 1: According to the editor’s suggestion, we edit our manuscript to meets PLOS ONE's style requirements, including those for file naming.

2: We suggest you thoroughly copyedit your manuscript for language usage, spelling, and grammar. If you do not know anyone who can help you do this, you may wish to consider employing a professional scientific editing service. 

Response 2: Thank you for your suggestions. I have used Editage to edit my manuscript for language usage, spelling, and grammar. The editing certificate is in the attachment.

3: Please note that PLOS ONE has specific guidelines on code sharing for submissions in which author-generated code underpins the findings in the manuscript. In these cases, all author-generated code must be made available without restrictions upon publication of the work. Please review our guidelines at https://journals.plos.org/plosone/s/materials-and-software-sharing#loc-sharing-code and ensure that your code is shared in a way that follows best practice and facilitates reproducibility and reuse.

Response 3: According to the editor’s suggestion, I have uploaded my code in https://gitee.com/Meliao/CAD.

4: We note that the grant information you provided in the ‘Funding Information’ and ‘Financial Disclosure’ sections do not match.

Response 4: Thank you for the reminder, ‘Financial Disclosure’ modify to the following: This work was supported by the Science and Technology Plan Project of the State Administration for Market Regulation, grant number 2019MK143.

5: We note that you have stated that you will provide repository information for your data at acceptance. Should your manuscript be accepted for publication, we will hold it until you provide the relevant accession numbers or DOIs necessary to access your data. If you wish to make changes to your Data Availability statement, please describe these changes in your cover letter and we will update your Data Availability statement to reflect the information you provide.

Response 5: According to the editor’s suggestion, I have added Data Availability statement in my manuscript.

6: We note that you have indicated that data from this study are available upon request. PLOS only allows data to be available upon request if there are legal or ethical restrictions on sharing data publicly. 

Response 6: According to the editor’s suggestion, I have uploaded data to the website https://gitee.com/Meliao/CAD.

7: Please review your reference list to ensure that it is complete and correct. If you have cited papers that have been retracted, please include the rationale for doing so in the manuscript text, or remove these references and replace them with relevant current references.

Response 7: According to the editor’s suggestion, I have updated my reference list.

Reviewer #1:

1: This study presented by the authors is well organized, the presentation of the article and the results are close to satisfactory. However, the reviewed literature studies are not up-to-date, the article should be reupload by examining the recent studies (especially 2020- 2021).

Response 1: Thank you for your suggestions. I have added the last 2 years of research in the introduction section.

 

---

## [Decision Letter · Decision Letter 1]

14 Apr 2022

Pressure vessel-oriented visual inspection method based on deep learning

PONE-D-22-00457R1

Dear Dr. Guixiong,

We’re pleased to inform you that your manuscript has been judged scientifically suitable for publication and will be formally accepted for publication once it meets all outstanding technical requirements.

Kind regards,

Antonio Riveiro Rodríguez, PhD

Academic Editor

PLOS ONE

Reviewers' comments:

Reviewer's Responses to Questions

**Comments to the Author**

1. If the authors have adequately addressed your comments raised in a previous round of review and you feel that this manuscript is now acceptable for publication, you may indicate that here to bypass the “Comments to the Author” section, enter your conflict of interest statement in the “Confidential to Editor” section, and submit your "Accept" recommendation.

Reviewer #1: All comments have been addressed

2. Is the manuscript technically sound, and do the data support the conclusions?

Reviewer #1: Yes

3. Has the statistical analysis been performed appropriately and rigorously? 

Reviewer #1: Yes

4. Have the authors made all data underlying the findings in their manuscript fully available?

Reviewer #1: Yes

5. Is the manuscript presented in an intelligible fashion and written in standard English?

Reviewer #1: Yes

6. Review Comments to the Author

Reviewer #1: I consider that the revisions have been carried out. this paper is acceptable for this valuable journal

7. PLOS authors have the option to publish the peer review history of their article (what does this mean?). If published, this will include your full peer review and any attached files.

Reviewer #1: No

---

## [Editor Report · Acceptance letter]

20 Apr 2022

PONE-D-22-00457R1 

Pressure vessel-oriented visual inspection method based on deep learning 

Dear Dr. Guixiong:

I'm pleased to inform you that your manuscript has been deemed suitable for publication in PLOS ONE. Congratulations! Your manuscript is now with our production department. 

Kind regards, 

on behalf of

Dr. Antonio Riveiro Rodríguez 

Academic Editor

PLOS ONE